# Exergy Analysis of a Bio-System: Soil–Plant Interaction

**DOI:** 10.3390/e23010003

**Published:** 2020-12-23

**Authors:** Masoomeh Bararzadeh Ledari, Yadollah Saboohi, Antonio Valero, Sara Azamian

**Affiliations:** 1Department of Energy Engineering of Sharif University of Technology, Tehran 11365-8639, Iran; bararzadeh@energy.sharif.edu (M.B.L.); azamian@energy.sharif.edu (S.A.); 2Department of Mechanical Engineering, University of Zaragoza, ETSII, 50015 Zaragoza, Spain; valero@unizar.es

**Keywords:** soil–plant system, exergy analysis, soil–plant exergy loss, soil–plant exergy destruction

## Abstract

This paper explains a thorough exergy analysis of the most important reactions in soil–plant interactions. Soil, which is a prime mover of gases, metals, structural crystals, and electrolytes, constantly resembles an electric field of charge and discharge. The second law of thermodynamics reflects the deterioration of resources through the destruction of exergy. In this study, we developed a new method to assess the exergy of soil and plant formation processes. Depending on the types of soil, one may assess the efficiency and degradation of resources by incorporating or using biomass storage. According to the results of this study, during different processes from the mineralization process to nutrient uptake by the plant, about 62.5% of the input exergy will be destroyed because of the soil solution reactions. Most of the exergy destruction occurs in the biota–atmosphere subsystem, especially in the photosynthesis reaction, due to its low efficiency (about 15%). Humus and protonation reactions, with 14% and 13% exergy destruction, respectively, are the most exergy destroying reactions. Respiratory, weathering, and reverse weathering reactions account for the lowest percentage of exergy destruction and less than one percent of total exergy destruction in the soil system. The total exergy yield of the soil system is estimated at about 37.45%.

## 1. Introduction

All ecosystem processes are irreversible and are accompanied by entropy production and exergy destruction. Exergy analysis is a method for analyzing energy systems based on the second law of thermodynamics. Exergy is a driving force for the conversion of energy and chemical substances for the further development of ecosystems [1]. Green plants convert exergy from the sunlight into exergy-rich biomass, via photosynthesis. The exergy content of biomass passes through different food chains in the ecosystems. At every trophic level, exergy is consumed and decomposing organisms dominate the last level in this food chain [2]. Most of the exergy is destroyed in the irreversible interaction of solar radiation with the surface, and only a tiny fraction of about 2.5% of global cosmic exergy consumption is used in transforming earth materials [3]. Exergy is stored in the form of biomass in soil. Increased inputs (more solar radiation is captured) mean more biomass, more exergy stored, and more exergy degraded [4]. Schneider and Kay [4] expressed an extended second law. The findings of ecosystem tend toward dissipating solar exergy because of different processes in the earth. They concluded that an increase in exergy storage and through-flow describe ecosystem development based on all three growth forms. The results referred to above can be used to formulate a general law for the development of ecosystems: if a system receives a through-flow of exergy (energy), it will move away from thermodynamic equilibrium and select the components and the organization that yield useful energy through the system (maximum power principles) and the most exergy stored in the system.

Ecological exergy analysis and environmental effects follow different approaches. Some researchers studied ecological integrity and the effects of environmental hazards. They focused on the effect of changing environmental conditions on the system, moving away from the original optimum operating point [5,6]. Some authors were interested in the information exergy and the eco-exergy methods. They analyzed ecosystems considering conservation [7] (Patten, Straškraba, and Jörgensen 1997), dissipation [8], openness [9], growth [10] (Sven E Jørgensen, Patten, and Straškraba 2000), constraints [11], and differentiation [12] toward an ecology of complex systems in a complex future [13]. They tried to find a method to illustrate ecosystems evolution considering the adjustment and compensation of human activity effects on the quality of ecosystems. This means the quality of ecosystems depends on the amount of exergy storage in the earth. Moreover, thermodynamic analysis of biological systems based on exergy analysis of photosynthesis [14], plant exergy efficiency [15], and exergy analysis of different biological reactions has been followed up. For instance, Lems [16] and Moura [17] demonstrated a method for biochemical reactions that can indicate a change in the quality of soil and plants. They introduced some methods for exergy analysis of different phenomena in the ecosystem by using exergy analysis of different reactions. In addition, according to Keller [18], the first and second laws of thermodynamics were used for a classical biological system (plant). Next, Petela [14] determined a system boundary of a leaf surface layer, in which biomass is created at temperature T and undertook an exergy balance of the leaf. Silva [15] decomposed the photosynthesis process into three main processes (photosynthesis I and II and Calvin cycle) and analyzed the changing quality of these processes. In this regard, Silow et al. [19] used and proposed the eco-exergy approach to analyze an open system that receives solar exergy as input, captures energy, as well as analyzes decomposer activities and the cycle’s waste, This method uses some coefficients, such as capacity of packaging information at the molecular level (DNA) that differs from one organism to another and can be taken into account using the eco-exergy function. This method can be applied to problems of a theoretical nature and does not have a practical application in ecosystems. Fath et al. [20] analyzed a bio-system including interactions between solar exergy, autotroph levels, and heterotroph levels. Their research used the eco-exergy method, considering different stages of growth and biomass storage. According to them, such analysis does not provide a better understanding of the transformation path [20]. In Ecoinvent Life Cycle Analysis, solar exergy absorbed directly by the soil is not considered by the functional unit, since its category is classified as “occupation, pasture and meadow, extensive”; this deficiency actually forgets the solar exergy the ecosystem needs to sustain its natural cycles [21]. Rocco et al. use exergy indicators for environmental impact assessment, which supplies a wider framework and deeper insights into the environmental performance of production processes and products. They used exergy-based indices for studying changes in soil quality [22].

Different types of nutrient loss and degradation reactions in ecosystems not only change the nutrient availability in the plant but also compromise soil microorganisms’ habitat. Moreover, the soil quality can decrease or increase due to nutrient uptake reactions such as mineralization by plants (microorganisms’ biochemical reactions in soil). In the present paper, we have utilized Moura and Lems’ exergy analysis in the plant–soil system [16].

In summary, there is a lack of systematic exergy analysis of the most important reactions in soil–plant interactions. Therefore, the primary objectives are:Evaluate the exergy performance of the bio-system (plant–soil system) by overall exergy efficiency.Identify the most significant source of exergy destruction and exergy losses in the bio-system (plant–soil system) and their location of occurrence.Evaluate the effect of various natural phenomena (weathering, acid rain, etc.) on the bio-system exergy efficiency.

## 2. Materials and Methods

We may consider soil as an energy system, in which the main purpose of this system is nutrient supply. As in any energy system consisting of different subsystems, in the soil system, numerous reactions occur that supply nutrients for plant growth [23].

Biogeochemical processes in the terrestrial environment dominate the hydrochemical response of small catchments because stream water is largely made up of drainage water from soils. Biogeochemical processes can be categorized into three major groups (cf. van Breemen et al., 1983):Biochemical processes, including interactions between biota and the atmosphere (e.g., photosynthesis, respiration, N2 fixation), and interactions between biota and soil solution (e.g., assimilation and mineralization).Geochemical and soil chemical processes, including interactions between solution and the soil solid phase (e.g., cation exchange, adsorption, chemical weathering).Chemical reactions in solution (e.g., hydrolysis, complexation reactions) or between solution and atmosphere (e.g., degassing of CO_2_).

In this study, these reactions are classified into five categories: plant to the soil, biota to the solution, atmosphere to biota, the solution to the atmosphere, solid to the solution, and other reactions (related to fertilizers, pollution, and acidification). The relationship between different parts of the ecosystems, including the hydrosphere, atmosphere, and biosphere, is shown in Figure 1.

The effects that lead to the generation of nutrients and increase their availability are considered positive and those that reduce the availability of nutrients and the degradation of nutrients in the soil are considered negative. However, the negative effects are those reactions that occur in the soil to compensate the entropy increasment; for example, in power plants, a condenser plays an important role to discharge the entropy increasment in the power plantsBy managing and controlling the soil system, these negative effects could be reduced to some extent, but cannot be eliminated (denitrification, de-complexation, and respiration). These interactions are associated with some main reactions including chemical formation, concentration change, electrical potential change, and mixing reactions. As mentioned, the main objective of the soil system is food production and the photosynthesis reaction. In order to achieve this goal, the main reactants of these reactions could come from minerals, manure/compost, and the atmosphere. The important elements, such as nitrogen, phosphorus, carbon, and sulfur, enter the soil from the atmosphere. Part of the reactions originate from the atmosphere to the soil or vice versa, which are considered to be positive, and reversing these reactions leads to the reduction in nutrients availability (negative effects). The most important part of the soil is the soil solution, which includes the organic and inorganic phases (Figure 2).

Variations in key electron donors and acceptors (${\mathrm{NO}}_{3}^{2-}$, $N_{2}$, ${\mathrm{NH}}_{4}^{+}$, ${\mathrm{SO}}_{4}^{2-}$, ${\mathrm{CH}}_{4}$, and dissolved organic carbon (DOC)) closely follow the predictions of thermodynamics. Transformations of N and other elements result from the response of microbial communities to two dominant hydrologic flow paths: (1) horizontal flow of shallow subsurface waters with high levels of electron donors (i.e., DOC, ${\mathrm{CH}}_{4}$, and ${\mathrm{NH}}_{4}^{+}$), and (2) near-stream vertical upwelling of deep subsurface waters with high levels of energetically favorable electron acceptors (i.e., ${\mathrm{NO}}_{3}^{2-}$, $N_{2}O$, and ${\mathrm{SO}}_{4}^{2-}$). Thermodynamic constraints on microbial metabolism depend on the use of electron donors and electron acceptors in redox reactions that generate energy for growth and maintenance. While organic matter (${\mathrm{CH}}_{2}O$) dominates as the electron donor in many natural environments, other electron donors (e.g., ${\mathrm{CH}}_{4}$, $H_{2}S$, Fe(II), ${\;\mathrm{NH}}_{4}^{+}$, and Mn(II)) can be locally important. Similarly, O2 dominates as the electron acceptor in oxic environments, while ${\mathrm{NO}}_{3}^{2-}$, $N_{2}O$, Mn(IV), Fe(III), ${\mathrm{SO}}_{4}^{2-}$, ${\mathrm{CO}}_{2}$, and ${\mathrm{CH}}_{2}O$ can be locally important in anoxic environments. Different combinations of electron donors and electron acceptors, in expression, release different amounts of free energy that, in turn, can be harnessed for microbial growth and maintenance. For example, aerobic respiration (${\mathrm{CH}}_{2}O$ as an electron donor, $O_{2}$ as an electron acceptor) generates almost five times more free energy (501 kJ) per mole of oxidized ${\mathrm{CH}}_{2}O$ than sulfate reduction does (102 kJ; ${\mathrm{CH}}_{2}O$ as electron donor and ${\mathrm{SO}}_{4}^{2-}$ as an electron acceptor) at pH 7 and molar concentrations of reactants [24]. The environment becomes increasingly reduced due to microbial consumption of electron acceptors following the sequence: (1) loss of $O_{2}$ (aerobic respiration); (2) loss of ${\mathrm{NO}}_{3}^{2-}$ (denitrification); (3) loss of ${\mathrm{SO}}_{4}^{2-}$ (sulfate reduction); and (4) accumulation of ${\mathrm{CH}}_{4}$ (methane fermentation) [24].

### 2.1. Exergy Analysis of Electron Transport Chain

Exergy transferred through an electron carrier is passed to the next carrier *i* − 1, used to do workwithin the living system, and partially lost to the environment as low-grade waste heat (exergy destruction). These carriers can move from the nucleus in channels to make energy carrieravailable for other reactions such as the carbon cycle (more detail about these reactions in photosynthesis process has been explained in the Appendix A) [15]:(1)$$B_{carriers,i}=B_{carriers,i-1}+W+\delta B$$ where *B_carriers_, _i_* is the exergy of carrier *i* (high-energy electrons proceed in photosynthesis reactions), *W* is the work performed by the electron transfer, and *δB* is the exergy destroyed. The standard reduction potentials can be expressed as:(2)$$\Delta G^{0}=-nF\Delta \varepsilon^{0}$$ where Δ*G*^0^ is the standard Gibbs free energy change (Here, the initial concentration of each component is 1.0 M, the pH is 7.0, the temperature is 25 C, and the pressure is 101.3 kPa.), *n* is the number of moles of electrons, *F* is the Faraday constant (96,485 Coulomb/mole e-), and Δ*ε*^0^ is the standard change in reduction potential. It can be modified to account for the effects of intracellular concentrations and used to calculate the exergy difference between electron carriers [16].

NADPH carries added protons in the last biosystem in photosynthesis processes. This process is a proton–electron that originated from the NADPH reaction. This reaction helps to reach a level of thermodynamic stability: $\Delta \varepsilon^{0}$ is 1.140 V [25]. The energy level difference compared to the reference responses of the intracellular proton–electron system in the exchange process is estimated as (detail of the reactions is presented in the Appendix A):(3)$$\Delta B_{elec}=B_{carriers,i}-B_{carriers,i-1}=nF\Delta \varepsilon^{0}+RT_{0}Ln(\prod {\left[A\right]}_{i}^{-\vartheta_{i}})$$ where Δ*B_elec_* is the exergy difference between carriers *i* and *i* ‒ 1, *R* is the universal gas constant (8.3143 J/mole- K), T0 is the dead-state temperature (298.15 K), ${\left[A\right]}_{i}$ is the activity of carrier *i*, and $\vartheta_{i}$ is the stoichiometric coefficient of carrier *i*.

For each molecule in the reactions, its chemical exergy is estimated using the method of Lems et al. [16]:(4)$$\begin{array}{c} B_{chem}\approx \sum_{k}(\vartheta_{k}B_{element,i})+\Delta G_{f}^{0}+RT_{0}Ln\left[A\right]+RT_{0}Ln\left(1+\sum_{i}\frac{\prod_{l=1}K_{l}}{{\left[H^{+}\right]}^{i}}\right)\\ +RT_{0}\sum_{j}Ln(1+\sum_{1}^{n}\prod_{i=1}K_{i}){\left[M_{j}\right]}^{i}) \end{array}$$ where *B_chem_* is the chemical exergy of a species (per mole), $B_{element,i}$ is the number of times that atom k occurs in the species (stoichiometric coefficient when forming the species from reference atoms), [*A*] is the activity of the species, $K_{l}$ is the chemical equilibrium constant (for either acid, base, or metal ion dissociation) for reaction *l*, ${\left[H\right]}^{+}$ is the hydrogen ion concentration, $\left[M_{j}\right]$ is the concentration of metal ion *j*, *k* is the atom counter, *i* and *l* are the reaction counters, and *j* is the metal ion counter [15].

To determine the exergy of a mole of photons, a modified form of Planck’s Law is applied [16]. Note that the only difference between Planck’s Law and the factor $\left(1-\frac{T_{earth}}{T_{sun}}\right)$, which accounts for a 5 percent difference between the energy and exergy of photons (the data required for this reaction are presented in the Appendix A):(5)$$B_{photon}\left(\lambda \right)=N_{A}\frac{hc}{\lambda }(1-\frac{T_{earth}}{T_{sun}})$$ where *B_photon_* is the photon exergy (J/mole photons) at a given wavelength (*λ*), *N_A_* is Avogadro’s number (6.023 × 1023), *h* is Planck’s constant (6.626 × 10 − 34 J × s), *c* is the speed of light (3 × 108 m/s), *λ* is the wavelength (m), T_earth_ is the ambient temperature of the earth (298.15 K), and *T_sun_* is the temperature of the sun’s surface (5762 K) [16].

Applying the mean-value theorem to exergy of the photon equation yields (the data required for this reaction are presented in the Appendix A):(6)$$B_{photon,avg}=N_{A}\;hc(1-\frac{T_{earth}}{T_{sun}})\frac{Ln\left(\lambda_{high}\right)-Ln\left(\lambda_{low}\right)}{\lambda_{high}-\lambda_{low}}$$

An exergy analysis of systems is the introduction of exergy cost analysis. This is based on the second law of thermodynamics regarding the concept of exergetic cost [26], the average cost approach [27,28], and specific exergy costing method [29]. In the present paper, we use the average exergy cost for bio-systems analysis.

### 2.2. The Exergy of Biochemical Reactions

In standard biochemical conditions, the medium is considered to be pure water of neutral acidity at standard pressure and temperature, and the compound is considered to be at unit concentration, or actually at the unit chemical activity. Gibbs free energies of formation in standard biochemical conditions have been determined for a wide range of biochemical compounds, and the exergy of these compounds at the given conditions can then be calculated following [16]:(7)$$Ex_{A}^{0^{\prime }}=\underset{i}{\sum }\vartheta_{i}Ex_{element,i}^{0}+\Delta_{f}G_{A}^{0^{\prime }}$$ where $Ex_{A}^{0^{\prime }}$ and $Ex_{i}^{0}$ are the stoichiometric number and the exergy in standard chemical conditions (superscript 0) of element *i* in compound A, respectively, and where $\Delta_{f}G_{A}^{0^{\prime }}$ is the Gibbs free energy of formation of compound A in standard biochemical conditions (superscript 0′). For the exergy of the elements, we refer to Szargut et al. (1988) [30], who also give a detailed description of how these exergy values are calculated.

The cellular concentrations of the compounds are far from the 1 M which is considered in Gibbs free energy, and since the environment can be considered water, the concept of an ideal solution can be applied, where the activity coefficient is considered to be equal to 1. Such an effect is very important as some reactions only occur due to differences in concentration.
(8)$$Ex_{concentration}=RT_{0}\ln\left[C/C_{0}\right]$$

The metabolic compounds suffer ionic dissociations with cations (H^+^) and anions (OH^−^), since the environment is active, forming different compounds, for example:$$C_{0}\underset{K_{1}}{\to }C_{1}+H^{+}\to^{K2}C_{2}+2H^{+}\ldots \to^{\mathrm{Kn}}C_{n}+{\mathrm{nH}}^{+}$$

The exergy due to acid dissociation is:(9)$$Ex_{dissociations}=-RT_{0}[1+\overset{n}{\underset{i=1}{\sum }}\frac{\prod_{j=1}^{i}Kj}{{\left[H^{+}\right]}^{i}}]$$

Some compounds bound to metallic ions forming metallic compounds and act as catalysts of the reactions. Similarly, an equation for the exergy effect of the metal compound is:(10)$$Ex_{metallic}=-RT_{0}ln[1+\overset{n}{\underset{i=1}{\sum }}(\overset{i}{\underset{j=1}{\prod }}Kj{\left[M\right]}^{i})]$$

The new equilibrium constant is known as the apparent equilibrium constant:(11)$$K_{apparent}=\underset{i}{\prod }{\left(Ci\gamma_{i}\right)}^{\vartheta_{i}}=\underset{i}{\prod }{\left(Ci\right)}^{\vartheta_{i}}\underset{i}{\prod }{\left(\gamma_{i}\right)}^{\vartheta_{i}}=K\Gamma$$ where γ is the activity coefficient and Γ is a global factor. The extended Debye–Hückel Law is used to evaluate Γ by re-evaluating the activity coefficient:(12)$$ln\gamma_{i}=-\frac{az_{i}^{2}I^{1/2}}{1+BI^{1/2}}$$ where *z_i_* is the ion electrical charge, *I* is the ionic force, and α and *B* are constants. The new equilibrium constants can now be re-evaluated using the definition of *pK = log*(k):(13)$$pK\left(I\right)=pK\left(I=0\right)-\alpha \log\left(e\right)\frac{I^{1/2}}{\left(I+BI^{\frac{1}{2}}\right)}\underset{i}{\sum }\vartheta_{i}z_{i}^{2}$$

The total exergy of the compound is then the sum of all effects above.
(14)$$Ex=Ex^{0}+Ex_{concentration}+\sum \left(Ex_{dissociations}\right)+Ex_{potential}$$

In some reactions, the proton and electron transfer can change the energy level of the product ($Ex_{potential})$. With the exergy value of the compounds evaluated, chemical reactions such as *A + B*→*C + D* can be analyzed through the exergy balance [17]:(15)$$Ex_{reactants}=Ex_{products}+Ex_{destroyed}+Ex_{lost}$$

Moreover, the exergetic efficiency can be defined by:(16)$$\varepsilon =\frac{Ex_{products}}{Ex_{reactants}}$$

## 3. Results

### 3.1. Exergy Loss and Destruction of Different Ecosystems Interactions

#### 3.1.1. Weathering

The weathering of rocks leads to the formation of sand, silt, and clay. Based on their mechanisms, the following are the three types of weathering: physical weathering, chemical weathering, and biological weathering. Chemical weathering (chemical erosion) is the decomposition of rocks by a change in the chemical and mineralogical composition, through a combination of several chemical processes. It is a slow but more intense process than physical weathering. Chemical weathering takes place mainly at the surface of rocks and minerals, leading to the disappearance of certain minerals and the formation of new products and secondary minerals. Chemical weathering is more intense in areas where it is preceded by physical weathering, which causes a decrease in particle size and an increase in surface area. Chemical weathering is therefore aided and abetted by physical weathering. During the process of chemical weathering, one or more of the following minerals in solution (cations and anions) are formed: oxides of iron and alumina (sesqui-oxides Al_2_O_3_, Fe_2_O_3_), various forms of silica (silicon oxide compounds), and stable wastes such as very fine silt (mostly fine quartz) and sand (coarser quartz) [31].

In sedimentary rocks, which are made up of primary and secondary minerals, weathering acts initially to destroy any relatively weak bonding agents (FeO), and the particles are freed and can be individually subjected to weathering.

Based on the results, the greater exergy destruction in chemical weathering is caused by hydrolysis reactions. Such exergy destruction is caused by the chemical reaction and the change in concentration (Table 1).

The main effects of these reactions in the soil are imbalanced creation in the number of cations and anions in the soil. Microorganisms have spent part of their activities on these reactions.

Therefore, it can easily reflect the effects of human activities on the soil balance reactions. Among these general reactions, chemical weathering reactions cause the 9% of total exergy destruction in the soil system.

#### 3.1.2. Dissolution and Precipitation

Solids can either be formed by precipitation and crystallization or dissolved depending upon the conditions of the solution. Some of the most influential conditions affecting dissolution/precipitation in soils are ionic composition and concentration, pH, and temperature. In addition, solution species that form strong complexes with the constituents of a solid may enhance the dissolution of such solids. In this section, we will discuss the process of solid phase formation and destruction [32].

Based on the thermodynamic prediction, mineral dissolution reactions are represented in Table 2.

As can be seen in Table 2, we only have some electrolysis reactions, which are dissolution reactions. These types of reactions are useful in the soil solution quality. The soil solution needs some ionic interactions to prepare nutrient that they have the proper condition to uptake by plant.

The most important precipitation reactions take place with $H_{2}{\mathrm{SO}}_{4}$ and ${\mathrm{HNO}}_{3}$. The exergy analysis of the reactions is shown in Table 3 [32].

The amount of exergy destruction in these reactions generally accounts for 1.43% of the total exergy destruction of the soil system.

#### 3.1.3. Soil Acidification and Leaching

Increasing amounts of acids can “mobilize” aluminum ions, which are normally present in an insoluble nontoxic form of aluminum hydroxide. It appears that when the soil pH dips to 5 or lower, aluminum ions are dissolved into the water and become toxic to plants. Aluminum ions cause the stunting of root growth and prevent the roots from taking up calcium. The result may be the overall slowing of the growth of the entire tree. Lower soil pH and aluminum mobilization can reduce populations of soil microorganisms. Soil bacteria have the job of breaking down the dead and decaying leaves and other debris on the forest floor. The effect of this action is to release nutrients such as calcium, magnesium, phosphate, nitrate, and others. Low pH and high aluminum ion concentrations inhibit this process. Higher amounts of acids can mobilize other toxic metals from the insoluble to the soluble ion forms in the same way as aluminum does it. The toxic metals include lead, mercury, zinc, copper, cadmium, chromium, manganese, and vanadium [33].

According to the results, the highest percentage of exergy loss due to acid rain is related to Al(OH)_3_ + H_2_SO_4_ and accounts for about 82% of total exergy destruction. The decomposition of H_2_CO_3_ reaction, CaCO_3_ + H_2_SO_4_→CaSO_4_ + H_2_CO_3_ reaction, and ion exchange reactions caused about 11%, 2%, and 5% of total exergy destruction, respectively.

In general, the amount of exergy destruction caused by acidification reactions is about 1% of the total exergy destruction of the soil system.

#### 3.1.4. Total Exergy Losses of Natural Processes in the Soil System

Exergy analysis is an applicable method for natural process loss estimation. This method enables us to present a new approach for natural resource depletion, especially in land resources. In this regard, Dewulf et al. [21] tried to establish a comprehensive resource-based life cycle impact assessment (LCIA) method that makes it possible to quantify the exergy taken from the natural ecosystems, and is thus, called cumulative exergy extraction from the natural environment (CEENE). Rocco et al. [22] utilized exergy life cycle analysis for different structures to counter the erosion losses of soil. He compared his approach with other methods such as Cumulative Exergy Demand (CExD), the Thermo-Ecological Cost (TEC), and the Cumulative Exergy Extraction from Natural Environment (CEENE), considering exergy requirements in different life cycle chains. The current study compares different soil erosion technologies considering their primary exergy requirement as well as the existence of their hidden impacts related to land use. In this paper, it is assumed that total exergy losses of natural processes in the soil system are caused by the negative effects of the soil system reactions. Furthermore, it should be mentioned that the exergy loss of different avoidable and unavoidable reactions in the soil can be used to estimate the total exergy losses (Table 4).

It is important to mention that some of the losses should be happening because they are critical for alive soil. Respiration is a sign of living biota in the soil, which is quite necessary for healthy soil, and the benefits overcome its unavoidable exergy losses.

Although some of the interactions are desirable reactions due to nutrient production, they are caused by material losses. Dissociation and acidification are the main sources of losses. This is because of their main role in metabolic reactions to interacting ionic materials.

#### 3.1.5. Exergy Loss and Exergy Destruction of Human Activity in the Soil

All ecosystems are open systems embedded in an environment from which they receive and discharge energy and matter. From a thermodynamic point of view, this principle is a prerequisite for the ecological processes. If ecosystems would be isolated, they would be at thermodynamic equilibrium without gradients and then, life. The openness explains, according to Prigogine, why the system can be maintained far from thermodynamic equilibrium without violating the second law of thermodynamics [19].

The soil nitrate pollution reaction is referred to as the reaction of nitrate with hydrogen cation (proton). Since nitrate ions are the critical material for nutrient uptake by plants and food production, this reaction reduces the concentration of nitrate in the soil; on the other hand, decreasing nitrites is a type of soil pollution called eutrophication. However, human activities have accelerated the rate and extent of eutrophication through both point-source discharges and non-point loadings of limiting nutrients, such as nitrogen and phosphorus, into aquatic ecosystems (i.e., agricultural eutrophication), with dramatic consequences for drinking water sources, fisheries, and recreational water bodies [34]. The high concentration of nitrate ions and the excessive use of chemical fertilizers are the main reasons for this phenomenon.

Heavy metal pollution is currently a major environmental problem because metal ions persist in the environment due to their non-degradable nature. The toxicity and bioaccumulation tendency of heavy metals in the environment is a serious threat to the health of living organisms. Unlike organic contaminants, heavy metals cannot be broken down by chemical or biological processes. Hence, they can only be transformed into less toxic species [35]. The toxicity of heavy metals in plants varies, depending on the plant species, specific metal involved, and concentration of metal, the chemical form of metal, and soil composition and pH. Metal toxicity is also shown in its ability to disrupt enzyme structures and functions by binding with thiol and protein groups, or by replacing co-factors in prosthetic groups of enzymes. Based on our results, lead and its compounds’ reactions have produced the most exergy destruction compared to other metals’ compounds. The lead compounds caused 1.87% exergy destruction of the soil. After lead, the exergy destruction originating from cadmium, copper, and zinc are, respectively, 1.55%, 1.14%, and 0.78% (Table 5). Note that exergy destruction is not a good indicator of toxicity, notwithstanding toxicity induces great effects on plants, microbiome, and life in general.

#### 3.1.6. Exergy Losses from Destructions in Processes of the Soil System

Exergy loss is due to undesirable reactions in each reactive group and is different from exergy destruction. For example, consumed exergy by weathering reactions is considered an exergy loss.

In the following figure, the amount of exergy destruction of soil processes is presented. Based on the results, acidification reactions due to acid rains lead to a greater amount of exergy destruction. Following acidification, the dissociation reactions (a reverse process of the protonation reaction) cause the second level of exergy destruction in degradation processes. Carbon cycle reactions and volatilization have the third and fourth places in this comparison (Figure 3).

The effects of natural process and human activity on exergy loss play a critical role in managing some desirable and undesirable reactions in the soil system. The detailed method for exergy analysis of these reactions is represented in the following section.

#### 3.1.7. Exergy Losses from Leaching Based on pH

Nutrient leaching is the downward movement of dissolved nutrients in the soil profile with percolating water. Nutrients that are leached below the rooting zone of the vegetation are at least temporarily lost from the system, although they may be recycled if roots grow deeper. Leached nutrients may contribute to groundwater contamination in regions with intensive agriculture. Nitrate leaching is also a significant source of soil acidification. In humid climates, some nutrient leaching occurs even under natural vegetation, but agricultural activities can greatly increase leaching losses [37]. According to the results, the total leaching exergy loss at different pH levels is shown in Table 6.

At pH 4 to 6, hydrogen, nitrate, and calcium ions are more active, and therefore, the greater amounts of leaching exergy losses have occurred in these pH levels.

#### 3.1.8. The Effects of Adding Fertilizers to the Soil

Fertilizers compensate for deficiencies when the nutrient concentration in the soil is less than required for plant growth. Considering the use of different types of fertilizers (organic or chemical), some reactions are added to the soil system. Although exergy destruction increases because of fertilizers, there is a tradeoff between exergy destruction caused by fertilizer reactions and nutrient deficiencies.

Generally, nitrogen, phosphorus, and potassium are the important nutrients that should be available for growth. Accordingly, these three types of fertilizers are added to the soil. The lack of nutrients in the soil is supplied through adding fertilizer/compost (similar to the energy carriers in energy systems). In uptake processes, plants use, in part, resources through mineralization. Another part of the resources compensates the losses occurred in the soil (Table 7; see the detailed reaction in the Appendix B).

Assuming that all of these fertilizers can be used in the final fertilizer, the phosphatic fertilizer-single superphosphate fertilizer compensates up to 48% of total exergy loss and destruction at pH (6 to 8) and therefore, it is the most effective fertilizer for compensating soil quality loss. Ammonium sulfate fertilizers and calcium ammonium nitrate have the least amount of efficiencies in compensating exergy loss and destruction (2 percent).

## 4. Discussion

### 4.1. Exergy Destructions Nutrient Supply for Plant Growth in the Soil System

Living organisms need energy to cover the maintenance of life processes. This energy is lost as heat to the environment in agreement with the second law of thermodynamics [19]. Thermodynamically, carbon-based life has a viability domain determined between about 250 and 350K. It is within this temperature range that there is a good balance between the opposing ordering and disordering processes: decomposition of organic matter and building of biochemical important compounds. At lower temperatures, the process rates become too slow and at higher temperatures, the enzymes catalyzing the biochemical formation processes will decompose too rapidly [19].

In the present paper, exergy loss and destruction are divided into avoidable and unavoidable. Avoidable exergy is related to some reactions that can be controlled by thermodynamics and biological conditions and parameters such as pH. Unavoidable exergy can be changeable in some conditions to maintain system stability (e.g., without respiration, photosynthesis will not improve). In Table 8, total avoidable and unavoidable exergy loss and destruction in bio-systems are observed.

The exergy destruction indicates the degree of irreversibility of the reactions. According to the classification, the exergy destruction of different paths is calculated (Table 9).

In Table 9, the exergy destruction of different paths is compared. According to this table, the most exergy destruction occurs in the biota atmosphere. One of the most important reactions in this pathway is photosynthesis, due to its low efficiency (about 15%) and the biota solution has the second greatest level of destruction. The main portion of exergy destruction in these processes is of a chemical nature.

In general, the highest percentage of exergy destruction of the soil system is related to photosynthesis reactions, which include photosystem I, photosystem II, and Calvin cycle reactions. After photosynthesis with 21% exergy destruction, humus and protonation reactions with 14% and 13% exergy destruction, respectively, are the most exergy destroying reactions. Respiratory, weathering, and reverse weathering reactions account for the lowest percentage of exergy destruction and less than one percent of total exergy destruction in the soil system.

As shown in Figure 4, the most exergy destruction in the biota–atmosphere relates to photosynthetic reactions, which account for 80% of total exergy destruction. Volatilization, carbon cycle, and ion exchange reactions account for 14%, 4%, and 2% of total biota–atmosphere exergy destruction, respectively.

Respiratory reactions result in slight exergy destruction and cause less than one percent of total exergy destruction in this group.

The general results of soil system reactions are represented in Figure 4. Based on the results, the total exergy yield of the soil system is reported at 37.45%.

### 4.2. Loss and Destruction of Exergy Due to Soil pH Levels in the Bio-System

Microorganism activities and ions involved in soils depend on its characteristics, especially soil pH. Different ions are activated at different pH levels [38].

According to the results, the least amount of exergy losses take place at pH = 2, in which weathering reactions cannot happen in the soil system. pH = 4 and pH = 6 have a greater amount of losses. At pH = 8, nitrate and sulfate ions have fewer activities and therefore, exergy losses will decrease (Figure 5).

At pH = 2, different solid and soil solution reactions cannot occur, and also, the greater amount of exergy destruction is associated with the biota–atmosphere subsystem. At pH = 4, solid and soil solution reactions are added to the system. However, at higher pH levels, the greatest destruction results from the biota solution. At pH = 8, the soil to plant reactions is omitted.

In the following figure, the exergy balance of the soil system can be observed. Generally, 191,127 kJ/mole of exergy input enters from solar energy, mineral resources, atmospheric resources, and fertilizer. In the neutral pH range (6 to 8), a significant amount of this exergy input can be lost as leaching and atmospheric influx, as well as different exergy destruction in soil. Eventually, only 71,568 kJ/mole of total exergy input will be available as a nutrient resource for the photosynthesis process (Figure 6).

## 5. Conclusions

Different interactions between soil, atmosphere, lithosphere, and biota can greatly affect the efficiency of exergy absorption from the sun and the amount of biomass exergy storage in the earth. Biological activities in soil supply the nutrients required by the plant; however, they could have unpleasant effects on microorganism life. In the present research work, the main biogeochemical reactions have been taken into account. The plant nutrients make up the inorganic elements required for plant growth, most of which are also essential for microflora and fauna to continue living. The main energy resources in soil include organic compounds—which are subject to biological attacks—serve as energy sources for the soil fauna and microflora rather than the few bacteria that can live. Although, these processes can form nitrogenous, phosphoric, and potassium compounds, they can cause about 17.8% of total exergy destruction in the soil. These organic and inorganic materials, after mineralization, are absorbed within ion exchanges. In general, different processes through mineralization to nutrient uptake destruct about 17.8% of total exergy input. A major amount of these resources are used by microorganism activities through biological reactions. Nutrient uptake involves different biological and chemical reactions, in which great portions of nutrients are lost into the atmosphere (similar to heat loss in energy systems). These reactions can deplete material resources in the soil through its interaction with water (equivalent to the discharge of waste and sewage in energy systems). In general, these losses are divided into two categories—losses of natural activities such as acid rain, erosion, etc., and the direct discharge of waste into the soil. Different biota–atmosphere reactions lead to high levels of exergy destruction in the soil system. One of the most important reactions in this pathway is photosynthesis (with a low amount of efficiency which is about 15%). After that, the next greatest amounts of exergy destruction occur in the biota–solution pathway. Given that, the most important reactions in this group force exergy destruction through chemical reactions. After photosynthesis, which accounts for up to 21% of total exergy destruction, the second and third places go to plant to soil reactions and protonation reactions with 14% and 13% of total exergy destruction, respectively. Respiration, weathering, and reverse weathering processes account for the lowest percentage of exergy destruction, with even less than one percent of total exergy destruction in the soil system. The lack of nutrients in the soil is supplied through adding fertilizer/compost (similar to energy carriers in energy systems). Plants use a part of resources through mineralization to uptake processes. Another part of resources compensates for the losses that occurred in the soil. In the neutral pH range (6 to 8), eventually, only 7.16E05 of 1.91E05 kJ/mole of the exergy input will be available as a nutrient resource for the photosynthesis process because the rest is lost due to leaching and atmospheric influx, as well as different exergy destruction in soil.

## Figures and Tables

**Figure 1 entropy-23-00003-f001:**
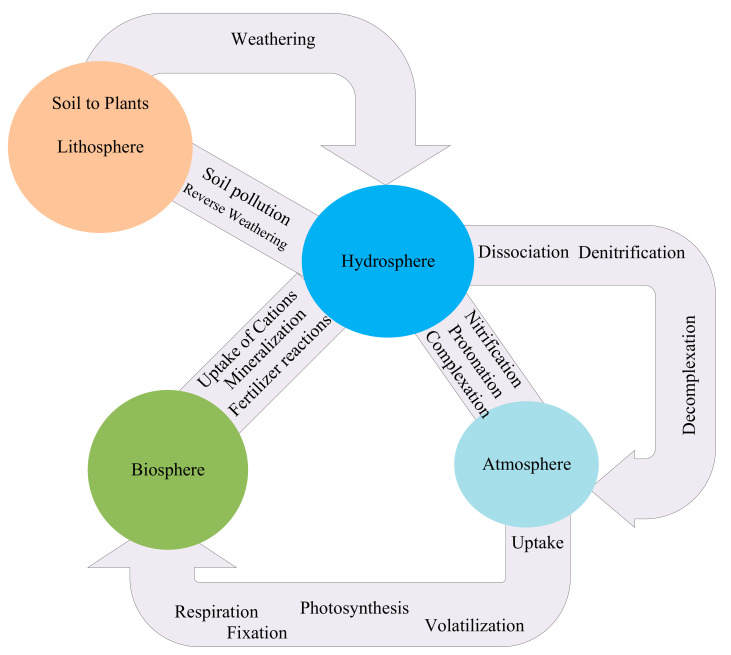
Mutual interactions between atmosphere, lithosphere, and biosphere leading to accretion or decrease in nutrient content in the soil.

**Figure 2 entropy-23-00003-f002:**
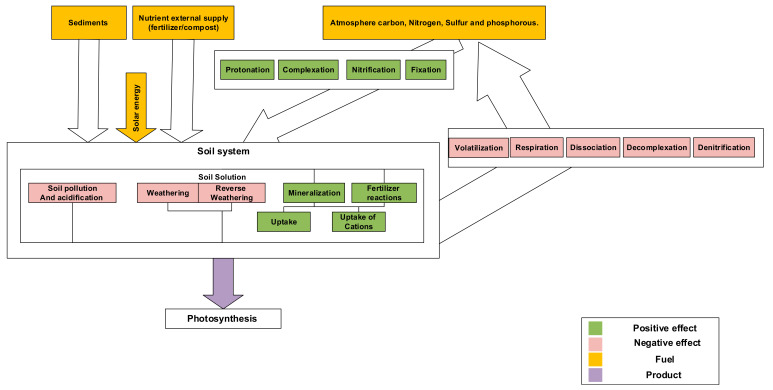
Soil–plant interactions.

**Figure 3 entropy-23-00003-f003:**
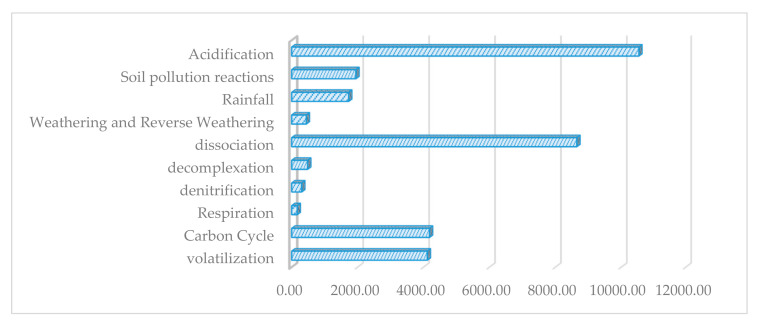
Exergy losses and destruction in reactions of the soil system (kJ/mole).

**Figure 4 entropy-23-00003-f004:**
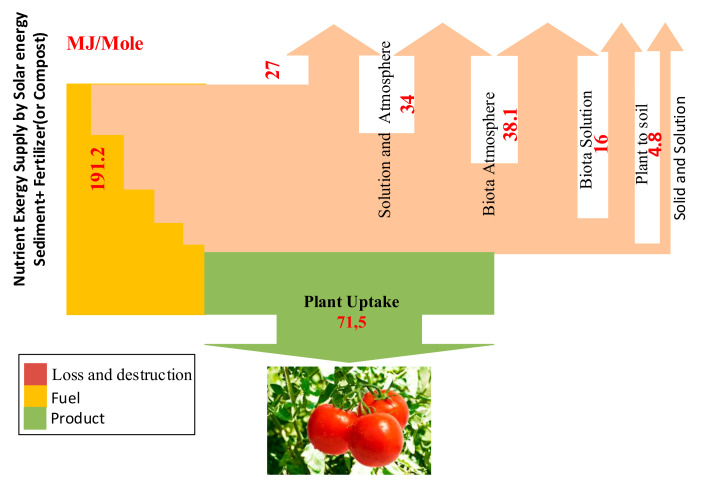
Sankey diagram of exergy analysis of nutrient supply for plant growth.

**Figure 5 entropy-23-00003-f005:**
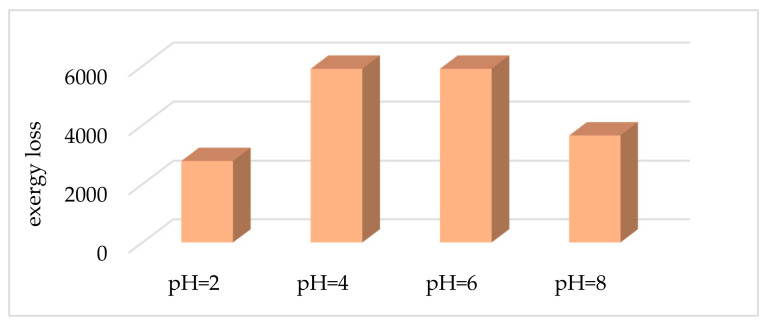
Soil exergy losses in different pH (kJ/mole of soil).

**Figure 6 entropy-23-00003-f006:**
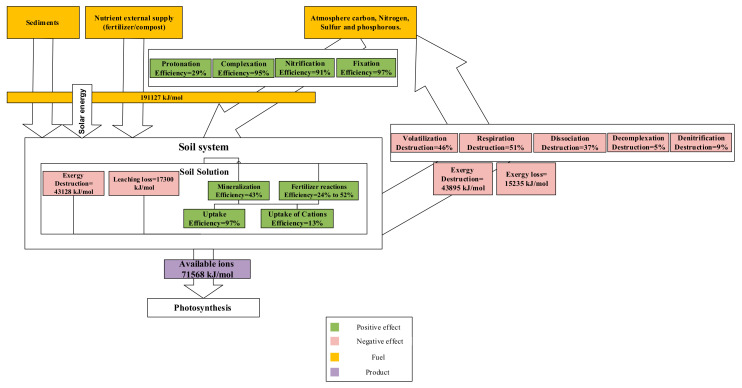
Different interaction of soil reactions (exergy efficiency and destruction).

**Table 1 entropy-23-00003-t001:** Exergy destruction of various chemical weathering (kJ/mole).

Reaction	$\Delta G{\;}^{1}\;(\mathrm{kJ}/\mathrm{mole})$	$Ex_{destruction}\;(\mathrm{KJ}/\mathrm{mole})$	$E_{in}{\;}^{2}\;(\mathrm{KJ}/\mathrm{mole})$
Oxidative weathering of mineral			
$4\mathrm{FeO}+O_{2}\to 2{\mathrm{Fe}}_{2}O_{3}$	−287.856	287.856	503.36
Reverse weathering			
${\mathrm{Fe}}^{2+}+\frac{1}{4}O_{2}+\frac{3}{2}H_{2}O\to \mathrm{Fe}{\left(\mathrm{OH}\right)}_{3}+2H^{+}$	151.92	171.59	376.59
$\mathrm{FeS}+\frac{9}{2}O_{2}+\frac{5}{2}H_{2}O\to \mathrm{Fe}{\left(\mathrm{OH}\right)}_{3}+{\mathrm{SO}}_{4}^{2-}+2H^{+}$	−960.22	73.19	393.47
Rainfall			
$2{\mathrm{Fe}}^{3+}+3{\mathrm{SO}}_{4}^{-}+6H_{2}O\to 2\mathrm{Fe}{\left(\mathrm{OH}\right)}_{3}+6H^{+}+3{\mathrm{SO}}_{4}^{2-}$	856.42	856.42	2582.8
Soil acidity adjustment			
${\left(H^{+}\right)}_{2}+{\mathrm{CaCO}}_{3}\to {\mathrm{Ca}}^{2++}+{\mathrm{CO}}_{2}+H_{2}O$	443	240.12	678.9

^1^ Difference of Gibbs free energy; ^2^
$Ex_{reactants}$.

**Table 2 entropy-23-00003-t002:** Thermodynamic analysis of mineral dissolution.

Reaction	log(K)	$Ex_{destruction}\;(\mathrm{kJ}/\mathrm{mole})$
${\mathrm{CaSO}}_{4}\left(\mathrm{gypsum}\right)↔{\mathrm{Ca}}^{2+}+{\mathrm{SO}}_{4}{}^{2-}$	−4.6	11.40
${\mathrm{CaCO}}_{3}\left(\mathrm{calcite}\right)↔{\mathrm{Ca}}^{2+}+{\mathrm{CO}}_{3}{}^{2-}$	−8.35	20.6
$\mathrm{Fe}{\left(\mathrm{OH}\right)}_{3}\left(\mathrm{amorphous}\right)↔{\mathrm{Fe}}^{3+}+3{\mathrm{OH}}^{-}$	−38.7	95.93
$\mathrm{Al}{\left(\mathrm{OH}\right)}_{3}\left(\mathrm{gibbsite}\right)↔{\mathrm{Al}}^{3+}+3{\mathrm{OH}}^{-}$	−33	81.80
$\mathrm{Fe}{\left(\mathrm{OH}\right)}_{2}H_{2}{\mathrm{PO}}_{4}\left(\mathrm{strengite}\right)↔{\mathrm{Fe}}^{3+}+2{\mathrm{OH}}^{-}+H_{2}{\mathrm{PO}}_{4}$	−35	86.75

**Table 3 entropy-23-00003-t003:** Thermodynamic analysis of precipitation.

Reaction	Log K	$Ex_{destruction}\;(\mathrm{KJ}/\mathrm{mole})$
$H_{2}{\mathrm{SO}}_{4}\to 2H^{+}+{\mathrm{SO}}_{4}^{2-}$	…	1157.18
${\mathrm{HNO}}_{3}\to H^{+}+{\mathrm{NO}}_{3}^{-}$	…	88.06
CaSO_4_↔Ca^2+^ + SO_4_^2−^	4.6	26.25
CaCO_3_↔Ca^2+^ + CO_3_^2−^	8.35	48.65
Fe(OH)_3↔_Fe^3+^ + 3 OH^−^	38.7	260.48
Al(OH)_3_↔Al^3+^ + 3 OH^−^	33.8	566.31
Al(OH)_2_H_2_PO_4_↔Al^3+^ + 2 OH^−^ + H_2_PO_4_	−30.5	174.08
Fe(OH)_2_H_2_PO_4_↔Fe^3+^ + 2 OH^−^ + H_2_PO_4_	−35	199.76
${\mathrm{Ca}}^{2+}+H_{2}{\mathrm{PO}}_{4}^{-}↔{\mathrm{CaHPO}}_{4}+H^{+}$	−0.08	1677.96
${\mathrm{Ca}}^{2+}+H_{2}{\mathrm{PO}}_{4}^{-1}↔{\mathrm{CaPO}}_{4}+2H^{+}$	−13.09	1752.18
	30.5	2778.64
$\mathrm{Al}{\left(\mathrm{OH}\right)}_{2}H_{2}{\mathrm{PO}}_{4}={\mathrm{Al}}^{3+}+2{\mathrm{OH}}^{-}+H_{2}{\mathrm{PO}}_{4}$ $\mathrm{Fe}{\left(\mathrm{OH}\right)}_{2}H_{2}{\mathrm{PO}}_{4}={\mathrm{Fe}}^{3+}+2{\mathrm{OH}}^{-}+H_{2}{\mathrm{PO}}_{4}$	35	2238.86

**Table 4 entropy-23-00003-t004:** The exergy loss of different avoidable and unavoidable reactions in soil.

Reaction	$Ex_{loss}\;(\mathrm{kJ}/\mathrm{mole})$
Volatilization	4111.76
Carbon Cycle	4177.53
Respiration	165.49
Denitrification	300.34
De-complexation	480.87
Dissociation	8648.41
Weathering and Reverse Weathering	440.17
Rainfall	1726.38
Soil pollution reactions	1946.24
Acidification	10,538.69
**Total**	**32,535.89**

**Table 5 entropy-23-00003-t005:** Exergy destruction of heavy metal reactions [36].

Element	$Ex_{destruction}\;(\mathrm{kJ}/\mathrm{mole})$	Total Exergy Destruction (kJ/mole)	Exergy Destruction Increased Percent
Pb	1628.3	88,651.2	1.87
Zn	680.2	87,703.1	0.78
Cd	1353.7	88,376.6	1.55
Cu	991.8	88,014.8	1.14

**Table 6 entropy-23-00003-t006:** The total exergy losses of leaching in different pH levels.

pH	$Ex_{loss}\;(\mathrm{kJ}/\mathrm{mole})$
2	2783.52
4	5904.97
6	5904.97
8	3649.42

**Table 7 entropy-23-00003-t007:** The exergy analysis of different fertilizer reactions.

Fertilizer	$Ex_{in}\;(\mathrm{kJ}/\mathrm{mole})$	$Ex_{out}\;(\mathrm{kJ}/\mathrm{mole})$	$Ex_{destruction}\;(\mathrm{kJ}/\mathrm{mole})$	$Ex_{loss}\;(\mathrm{kJ}/\mathrm{mole})$	Efficiency (%)
Urea	20,761.7	8716.3	11,933.4	112.0	42.0
Ammonium Sulphate	5617.9	1389.2	4228.8	0.0	24.7
Calcium Ammonium Nitrate	5972.8	1903.9	4068.9	0.0	31.9
Phosphatic fertilizer	29,323.40	15,001.68	13,862.95	458.77	51.2
Phosphatic fertilizer-single Super Phosphate	160,320.3	54,736.8	100,995.8	4587.7	34.1

**Table 8 entropy-23-00003-t008:** Total avoidable and unavoidable exergy loss and destruction in bio-systems.

$Ex_{AExD}{\;}^{1}\;(\mathrm{kJ}/\mathrm{mole})$	$Ex_{UAExD}{\;}^{2}\;(\mathrm{kJ}/\mathrm{mole})$	$Ex_{AExL}{\;}^{3}\;(\mathrm{kJ}/\mathrm{mole})$	$Ex_{UAExL}{\;}^{4}\;(\mathrm{kJ}/\mathrm{mole})$	Total (kJ/mole)
29,252.81	51,840.28	23,299.9	9236	113,629

^1^ Avoidable exergy destruction. ^2^ Unavoidable exergy destruction. ^3^ Avoidable exergy loss. ^4^ Unavoidable exergy loss.

**Table 9 entropy-23-00003-t009:** The overall exergy analysis of the soil system.

Processes	$Ex_{in}\;(\mathrm{kJ}/\mathrm{mole})$	$Ex_{out}\;(\mathrm{kJ}/\mathrm{mole})$	$Ex_{destruction}\;(\mathrm{kJ}/\mathrm{mole})$	$Ex_{loss}\;(\mathrm{kJ}/\mathrm{mole})$	Efficiency (%)
Plant to Soil	19,133.50	4099.82	13,307.31	1726.38	21.43
Biota Solution	67,758.59	29,781.03	25,492.62	12,484.94	43.95
Biota Atmosphere	51,194.08	17,173.84	25,565.45	8454.78	33.55
Solution and Atmosphere	45,925.62	18,296.86	18,199.14	9429.62	39.84
Solid and Solution	5038.48	270.28	4328.03	440.17	5.36
total	191,126.90	71,568.07	32,535.89	87,022.94	37.45

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
