# Peer review of "Exergy Analysis of a Bio-System: Soil–Plant Interaction"

_entropy, 2020, doi:10.3390/e23010003_

Round 1
Reviewer 1 Report
I think the article will be in the low future a reference in the Exergy analysis of living beings. There are some issues that we would like to discuss in order to the article improve its quality. I congratulate the authors. My major reviews are with the intent to improve even more an excellent article,
One minor point is that figure 6 is with a reference to other journal of Mdpi energies, and all equations are blurry , it must be corrected. It seems a problems of the typing software....
The Exergy of biochemical reactions authors claim that the use a biochemical reference, vender the less the work of the professor lema cited in this article was to modify the references to the actual condition of the plant.
one issue that was not waisted in the article, is th ATP formation, which is a bound of organic and inorganic things, how authors evaluated. This item in the text, the biochemical reactions seems to be based on less model, and I think this is the only point in which the authors work did not explain accordingly in there thesis and articles. Authors could open this model for us and make a critics analysis .
In table three three are two identical equation involving aluminium ...
one interesting pint that authors account for the lost of quality of energy in mineralization of the plant, but what about soil Exergy destruction? And it is not clear if the destroyed Exergy is from the reaction of the fertilizers described in the table a23.
one point is again the atp reaction, chemical compound that exists only in biological systems , but with a bound of organic and inorganic elements. How does authors explain this.
Reviewer 2 Report
- The work is difficult to follow, the calculations are presented without a clear explanation of their origin or of the improvement percentages that are presented. There are several formatting errors in text, tables, and in general in the presentation of the work.
- Grammar and English revision are suggested, the text has several typing errors and inconsistencies in subscripts used in chemical formulas or equations. Review and improve.
- In equation 1, what does the subscript i-1 mean? What position is concerned? Explain in the manuscript.
- The use of equations (1-3) predisposes a series of hypotheses around an ideal gas and other considerations, please list or describe them before developing the models.
- Table 1, it is not clear, I understand that 5 different reactions are presented with their respective results, but the term “soil acidity adjustment” presents two lines of results.
- Page 10, 12, and 16. The sentence “2Error! Reference source not found..” it's in the manuscript.
- Table 2. Incorrect use of subscripts in chemical formulas.
- Page 10. According to the authors “Table 2 shows it as a general case since 0.34% of total exergy destruction is related to mineral dissolution” How do you calculate this percentage? And which calculation from Table 2 refers to, please clarify.
- Page 12. The sentence “This is a desirable loss” it is in the manuscript; the idea is not understood.
- Page 15. In agreement with the authors “In the present paper, exergy loss and destruction are divided into avoidable and unavoidable. Avoidable exergy is related to some reactions that can be controlled by thermodynamics and biological conditions and parameters such as pH”. However, this concept could be discussed with the one presented by the workaround of the Advanced Exergy Analysis by the research group of Prof. Tsatsaronis, where the avoidable part of the exergetic destruction is attributed to what can be reduced and the unavoidable part to what cannot be reduced. It gives me the idea of not being the same concept.
- Page 17. The sentence “Error! Reference source not found..” this is in the manuscript.
- Page 19. Image from another magazine "energies" is in the manuscript. Perhaps it comes from another work, which is not well referenced in the figure or text
For all this, I do not recommend the work to be considered in Entropy
Reviewer 3 Report
In this paper, a thermodynamic analysis has been investigated to compare the most important reactions in the soil plant interactions in terms of exergy destruction/efficiency. The authors discussed a significant number of parameters’ effects on the exergy destruction developing a new method. In the reviewer’s opinion, this paper will contribute to the relating literature, if following major recommendations are taken into considerations;
Although introduction is very informative, it could be improved. Some references are missing. For example, Silow et al. [?], Dewulf et al. [?], Rocco et al. [?]
- The reviewer thinks that the findings in the paper have not been discussed in detail, and expect more discussions, especially in Section 3. For example, in sections 3.1.1 and 3.1.2, although there is a detailed background in the beginning, the result discussion is only two sentences.
- Exergy destruction is one of the key factors in this paper. However, there is no information how exergy destruction (Exlost) is calculated? It is same for Exponential.
- The language and style should be improved.
Minor Issues in the paper are as follow:
- In table 1: What is Ein? It was not defined in the equations.
- Section 4.1: There are repeated sentences.
- Figure 5: What the y – axis represent?
- Please, do not use images for equations. Word – equation editor could be used.
- Definition of some symbols are repeated after each equation. For example, ΔG, R, To, were defined just after both Eqns. (3) and (4). Lasly, after Eqn. (3) to should be To.

Round 2
Reviewer 1 Report
I think the article may be accepted as is. Nevertheless, the reviewer is a little confused regarding the supplementary material, the .docx sent to me. Is it going to be on the article? That´s the sole reason why I asked for minor reviews.
The rest of the manuscript now is in accordance with the journal standards. I think it is suitable for publication
Reviewer 2 Report
The authors have responded in a good way to my comments, they reviewed the grammar (mostly in articles and prepositions), as well as, my doubts have been clarified.
Reviewer 3 Report
The reviewer thinks that the quality of the paper is improved by adding more discussion to the result and discussion sections.